# Optimizing glycine concentration to enhance gibbsite-catalyzed abiotic humification of catechol and glucose

**Kai Li, Qi Han, Jingjing Wang, Donghui Dai, Haoyu Gao, Jingwei Gao, Mingshuo Wang, Haihang Sun, Shuai Wang** [ID] *

College of Agriculture, Jilin Agricultural Science and Technology College, Jilin, China

* wangshuai@jlnku.edu.cn

## Abstract

The Maillard reaction represents a pivotal biochemical pathway for the abiotic formation of humic-like substances (HLSs); however, the regulatory role of gibbsite ($\alpha$-Al(OH)$_3$) in mediating this process remains insufficiently explored. This study systematically evaluated the effects of glycine concentration (0–0.24 mol/L) on the abiotic humification of catechol (0.06 mol/L) and glucose (0.06 mol/L) in the presence of gibbsite, using a sterile liquid shake-flask incubation system. The molecular complexity of the supernatant, total organic carbon (TOC) retention efficiency, and structural evolution of HLSs isolated from the dark-brown residue were analyzed through UV-Vis spectroscopy, TOC quantification, Fourier-transform infrared (FTIR) spectroscopy, and elemental analysis. Results demonstrated that: (1) The Gly0.24 treatment (0.24 mol/L glycine) achieved the minimal TOC loss, with a reduction of only 26.0% compared to 49.8% in the control group (without glycine). This carbon-preserving effect was attributed to the formation of Al–C complexes. (2) At a glycine concentration of 0.12 mol/L, the resulting HLSs exhibited the highest degree of aromatic condensation—evidenced by the lowest $E_4/E_6$ ratio (2.11)—and the richest content of oxygen–containing functional groups (O/C atomic ratio = 1.38). Concurrently, FTIR analysis indicated suppressed vibration of Al–O bonds in this treatment, suggesting that moderate glycine concentrations could modulate gibbsite–organic interactions to favor humification. (3) The Gly0 (no glycine) and Gly0.06 (0.06 mol/L glycine) treatments yielded the maximum humic-like acid (HLA) content, with respective increases of 1295.9% and 1034.6% relative to the control. This observation implies that low glycine levels (or its absence) primarily promoted the polymerization of catechol and glucose into HLA, rather than diverting carbon toward other reaction products. (4) Higher glycine concentrations (0.12–0.24 mol/L) significantly enhanced the accumulation of nitrogen-containing compounds in HLA, leading to a marked decrease in the C/N ratio (down to 8.7 in Gly0.24). This trend confirmed that excess glycine served as a nitrogen donor, facilitating the incorporation of nitrogen moieties into HLA structures

**Data availability statement:** All relevant data are within the manuscript and its Supporting information files.

**Funding:** Science and Technology Project of the Education Department of Jilin Province, China (grant number JJKH20240504HT). The funders had no role in study design, data collection and analysis, decision to publish, or preparation of the manuscript.

**Competing interests:** The authors have declared that no competing interests exist.

during humification. These findings highlighted that 0.12 mol/L glycine represented the optimal concentration for optimizing abiotic humification in the gibbsite system, as it balances two critical processes: aromatic polycondensation (a hallmark of humification degree) and the enrichment of oxygen-containing functional groups (key for HLS reactivity). This study provided novel mechanistic insights into gibbsite-catalyzed Maillard pathways, thereby advancing the development of strategies for efficient carbon sequestration in terrestrial ecosystems and the valorization of lignin-rich agricultural/industrial wastes into high-value humic-based products.

## 1. Introduction

Amino acids (AAs) and amino sugar residues in soils are crucial sources of nitrogen [1]. AAs described in lignin/phenol-protein theory and in the Maillard reaction are considered as important precursors in the formation of humic substances (HSs) via oxidative polymerization processes, which take place in soils or during composting [2]. The quantitative significance of Maillard reaction under specific environmental conditions has been a topic of contention throughout the history of HSs chemistry, given that the optimal conditions for these reactions are not prevalent in most ecosystems. Based on the theory of HSs formation, AAs play important roles in HSs formation. Serving as essential nutrients for microbial activity, AAs are widely regarded as primary contributors to HSs formation through oxidative polymerization during the Maillard reaction during composting or in the lignin/phenol-protein theory [3,4].

According to Ma et al. [5], AAs-derived ionic liquids can significantly facilitate lignocellulose degradation and humification. Incorporating N sources in the composting of plant-derived waste is an effective approach for boosting organic matter degradation and accelerating the humification process. In addition, the enhancement of lignocellulose-like biomass composting through humification processes might be closely associated with precursors interactions [6]. The introduction of exogenous AAs or their ionic liquids was shown to improve lignocellulose degradation and HSs synthesis during straw biomass composting [7,8]. The addition of phenylalanine and leucine not only altered bacterial community functions but also directly contributed to HSs synthesis as precursors, thereby promoting compost humification [9]. However, HSs are believed to form through biotic and abiotic pathways [10]; these products formed via abiotic processes are termed humic-like substances (HLSs). Abiotic pathways often neglected in existing composting or theoretical studies on HLSs formation, typically exclude microbial involvement [11]. Among these reactions, only the Maillard reaction specifically involves the polymerization of reducing sugars with N compounds without microbial participation. Recognized as a nonenzymatic browning process, the Maillard reaction is also an abiotic humification pathway catalyzed by $\delta$-$MnO_2$, using glucose, glycine, and catechol as precursors, which are crucial and highly reactive components affecting HLSs formation through the Maillard pathway [2]. Mu et al. [12] clarified that the Maillard reaction, which did not involve microorganisms, played a significant role in

the formation of humic acid (HA) in the aerobic fermentation of cow dung, chicken manure, and rice straw, suggesting a predominance of abiotic pathways in HA formation. This reaction entails the condensation of carbonyl amines to form HA and can occur spontaneously through heating, independent of microorganisms, biological enzymes, or chemical reagents. It involves the cleavage, rearrangement, and polymerization of proteins and sugars to form N-containing heterocycles known as HA. Zou et al. [13] examined the specific roles of $MnO_2$ in HS formation through the oxidative polymerization of catechin and glycine, where $MnO_2$ primarily served as a catalyst and an oxidant, significantly enhancing HA formation. Humification is markedly improved by metallic oxides in nature. Humic-like acids (HLA) are organic macromolecules with abundant strong binding sites, such as phenolic –OH and –COOH groups [14]. The formation of HLAs primarily involves biological transformations and chemical processes [15]. Natural HSs are predominantly produced via biotic humification processes [16]. However, abiotic processes have garnered considerable attention for their rapidity and efficiency, and the structural properties of HLAs can be manipulated by altering reaction conditions [17,18]. Metal oxides (Mn/Fe/Al/Si oxides) are key inorganic mineral components that enhance HLAs formation, with Fe(III) and Mn(IV) oxides being the most extensively studied [19]. In contrast, Al(III) (hydr)oxides—despite their abundance in soils and potential as Lewis acid catalysts—have received limited attention. Research on Fe(III) and Mn(IV) oxides was more prevalent than on other oxides, primarily due to their strong redox activity (e.g., Mn(IV)/Mn(II) and Fe(III)/Fe(II) cycles) that directly drove oxidative polymerization of humic precursors [13,19]. Unlike Mn/Fe oxides that primarily acted as oxidants to drive precursor polymerization [13], gibbsite's layered structure provided abundant surface –OH groups that facilitated nucleophilic additions between Maillard precursors (e.g., glycine, catechol), while its point of zero charge (PZC = 8.2) enabled stable interactions with both anionic (e.g., carboxylates) and neutral (e.g., phenols) organic molecules [1]. This unique combination of structural and surface properties suggests gibbsite may exhibit a non-oxidative catalytic mechanism in abiotic humification, yet its synergy with amino acids (e.g., glycine) as both N-containing precursors and reaction regulators has not been systematically investigated.

Given the potential of exogenous precursors to enhance the humification process during composting, their proportional relationship to HLSs formation has emerged as a significant area of research [20]. Zhang et al. [21] investigated this by adding varying concentrations of catechol to systems containing glucose and glycine, aiming to elucidate the abiotic humification mechanism by monitoring the fate of these precursors. Their findings revealed that higher catechol concentrations could accelerate the formation of fulvic-like acids (FLA) and HLA, and increase the degree of unsaturation in HLA. Similarly, Hardie et al. [22] demonstrated that increasing the molar ratio of glucose to catechol and glycine in a catechol-Maillard system enhanced the production of low-molecular-weight, strongly aliphatic carboxylic Maillard reaction products in the supernatant, analogous to natural HA. Nonetheless, the abiotic effects of AAs have been infrequently explored. Considering that precursors significantly influence HS formation [23], changes in the ratio of Maillard precursors inevitably affect the characteristics and composition of HLSs. Notably, while previous studies have optimized precursor ratios (e.g., catechol:glucose) for Mn/Fe oxide-catalyzed humification [21,22], no study has explored how glycine concentration—a key N source in Maillard reactions—modulates gibbsite-mediated abiotic humification. Therefore, the liquid shake-flask culture method was employed. Fixed concentrations of catechol and glucose solutions were inoculated into a phosphate buffer containing gibbsite, varying only the glycine concentration in the sterile culture system. The supernatant and dark-brown residue were collected via centrifugation. The $E_4/E_6$ ratio and total organic C (TOC) content of the supernatant, the humus composition and FTIR spectra of the dark-brown residue, and the elemental composition of HLA extracted from the dark-brown residue were analyzed to determine the optimal concentration of exogenous glycine for the abiotic humification of catechol and glucose as mediated through the Maillard pathway involving gibbsite. The primary objectives of this study were: (1) to systematically evaluate the effect of glycine concentration (0–0.24 mol/L) on the abiotic humification of catechol (0.06 mol/L) and glucose (0.06 mol/L) in a gibbsite-catalyzed sterile system; (2) to characterize changes in TOC retention, molecular complexity of supernatants, and structural evolution of HLSs using UV-Vis, TOC quantification, FTIR, and elemental analysis; (3) to identify the optimal glycine concentration that balanced aromatic polycondensation and

functional group enrichment; and (4) to provide insights for optimizing C sequestration and valorizing lignin-rich agricultural waste via gibbsite-mediated Maillard reactions.

## 2. Materials and methods

### 2.1. Preparation of materials

Gibbsite ($\alpha$-Al(OH)$_3$) was synthesized according to Rosenqvist et al. [24] to ensure controllable purity and physicochemical properties: Briefly, a 1.0 mol/L AlCl$_3$ solution was added dropwise to a 4.0 mol/L NaOH solution with continuous stirring until the pH reached 4.6, resulting in the formation of amorphous aluminum hydroxide. The resulting suspension was then aged at 40°C for 2 hours, electrodialyzed with ultrapure water for 3 months at room temperature using a bipolar membrane stack (cation exchange membrane: Nafion 117, DuPont; anion exchange membrane: AMX, Tokuyama) at a constant current density of 15 mA/cm$^2$ and a voltage of 8 V to remove residual Cl$^-$ and Na$^+$ ions. Subsequently, the product was dried at 60°C, ground, and sieved through a 149 µm mesh. The reason for choosing artificially synthesized gibbsite was that commercially available gibbsite often contains trace impurities (e.g., Fe/Mn oxides) that could interfere with abiotic humification [24]. The synthesized gibbsite was characterized for purity: specific surface area (SSA = 21.85 m$^2$/g) and point of zero charge (PZC = 8.2) matched pure gibbsite standards [24]; FTIR spectroscopy (Fig 5) showed a sharp Al–O lattice vibration peak at 611–656 cm$^{-1}$ without impurity bands. Analytical grade catechol (C$_6$H$_6$O$_2$), glucose (C$_6$H$_{12}$O$_6$) and glycine (C$_2$H$_5$NO$_2$) were sourced from Sinopharm Chemical Reagent Co., Ltd., China. Preparation of 0.2 mol/L phosphate buffer (pH 8.0): 5.3 ml of 0.2 mol/L NaH$_2$PO$_4$ and 94.7 ml of 0.2 mol/L Na$_2$HPO$_4$ were thoroughly mixed, and 0.02% thimerosal (C$_9$H$_9$HgNaO$_2$S) was added. This pH was selected for two key reasons: (1) it was close to gibbsite's point of zero charge (PZC = 8.2), minimizing non-specific electrostatic adsorption of Maillard precursors (e.g., glycine, catechol) onto gibbsite surfaces and ensuring the catalyst's activity was focused on mediating humification reactions [1]; (2) pH 8.0 simulated the neutral-to-weakly alkaline conditions of composting microenvironments or agricultural soils, enhancing the ecological relevance of our results [6,20].

### 2.2. Experimental method

#### 2.2.1. Experimental design.
Sterile conditions were maintained throughout the experiments to ensure predominance of abiotic transformation. All glassware, phosphate buffers, and other equipment were sterilized through autoclaving before use. Five hundred-milliliter conical flasks were used for the experiments. Each flask contained 250 ml of 0.2 mol/L phosphate buffer, into which 2 g of gibbsite were added. Both catechol and glucose were introduced at 0.06 mol/L (consistent with Zhang et al. [21], who showed this ratio enhanced HLS formation in catechol-Maillard systems) and incubated with various concentrations of glycine (0, 0.03, 0.06, 0.12, and 0.24 mol/L), denoted as Gly0, Gly0.03, Gly0.06, Gly0.12 and Gly0.24, respectively—a concentration range referenced from Ma et al. [5] to cover low-to-high amino acid inputs and identify optimal humification conditions. Hardie et al. [22] further confirmed that ~0.06 mol/L glucose-catechol mixtures produced HLSs analogous to natural HA. The control (CK) consisted of 2 g of gibbsite added to the phosphate buffer. All reactions were conducted in triplicate. Reproducibility was validated by calculating the coefficient of variation (CV) for key parameters (e.g., TOC content, $E_4/E_6$ ratio) across replicates; CV values were consistently < 5%, indicating high experimental precision [21].

The liquid shake-flask culture was initiated under a sterile and dark-controlled environment at 28°C—consistent with the average temperature of temperate agricultural soils [11], avoiding high-temperature decomposition (>40°C) or low-temperature kinetic inhibition (<20°C)—with a rotation speed of 150 rpm. Throughout the culture period, a 2 ml aliquot of the supernatant was extracted at predetermined times (0, 3, 6, 18, 28, 48, 76, 124, 172, 240, and 360 hours) based on pre-experimental kinetics: (1) Early stages (0–18 h) to capture rapid precursor cleavage and initial polymerization; (2) Middle stages (28–124 h) to monitor the accumulation of intermediate products; (3) Late stages (172–360 h) to track humification maturation [21]. Each aliquot was centrifuged at 16,000 rpm for 5 minutes, and 1 ml of the supernatant was diluted

to 25 ml for UV-Vis analysis [$E_4/E_6$ ratio was determined from absorbance measurements at 465 nm ($E_4$) and 665 nm ($E_6$), respectively, by calculating the ratio of the two absorbance values]. TOC content was measured using a Vario TOC cube analyzer (Elementar, Germany).

**2.2.2. Extraction of humic-like acid (HLA).** At 0, 18, 48, 76, 124, 172, 240, and 360 h, a 10 ml aliquot of supernatant was withdrawn, centrifuged at 16,000 rpm for 10 min, and the dark-brown residue was acidified to pH 1.0 with 1.0 mol/L HCl (equilibrated for 24 h). Complete HLA precipitation was confirmed by two steps: (1) no additional flocculation was observed when the supernatant was re-acidified to pH 0.8; (2) UV-Vis spectroscopy showed no absorbance at 280 nm (a characteristic wavelength for HLA) in the post-centrifugation supernatant. Following confirmation, the mixture was subjected to another high-speed centrifugation (16,000 rpm for 10 minutes): the clear supernatant (fulvic-like acid, FLA) was neutralized and diluted to 25 ml; the precipitate (HLA) was dissolved in 0.1 mol/L NaOH at 60°C, neutralized, and diluted to 25 ml. C contents of HLA and FLA ($C_{HLA}$ and $C_{FLA}$) were measured via TOC analysis. After 360 h, remaining HLA was purified by rinsing with 150 ml of 6% HCl-6% HF to remove inorganic impurities, magnetically stirred for 24 h, centrifuged (16,000 rpm for 10 min), freeze-dried, ground, and sieved through a 0.01 mm mesh for FTIR and elemental analysis.

## 2.3. Chemical analysis

The absorbances of diluted aliquots of the supernatant and diluted liquid HLA sample were determined at 465 ($E_4$) and 665 nm ($E_6$) using a UV-visible spectrophotometer (TU-1900, Beijing Purkinje General Instrument Co., Ltd., Beijing, China). Subsequently, the $C_{HLA}/C_{FLA}$ ratio was calculated. The elemental compositions of C, H, O, and N in solid HLA samples were determined using an elemental analyzer (PE 2400II CHNS/O, Perkin-Elmer, Waltham, MA, USA). For further structural characterization of the dark-brown residue, Fourier transform infrared (FTIR) spectrophotometry was performed using an FTIR spectrometer (FTIR-850, Tianjin Gangdong Sci & Tech Development Co., Ltd., Tianjin, China). FTIR spectra, covering a wavenumber range of 400–4000 cm$^{-1}$, were recorded. The acquired spectra were analyzed using FTIR 850 software and visualized using Origin 20.0 software (OriginLab Corporation, Northampton, MA, USA).

## 2.4. Statistical analysis of the data

Data analysis and spectral processing were performed using Microsoft Excel 2003 and Origin 20.0 software, respectively. Statistical analysis was conducted via SPSS 18.0, with one-way analysis of variance (ANOVA) used to evaluate overall group differences, followed by the least significant difference (LSD) test for post-hoc pairwise comparisons.

## 3. Results

### 3.1. $E_4/E_6$ ratio and total organic carbon (TOC) in the supernatant

The $E_4/E_6$ ratio serves as an optical indicator that reflects the molecular complexity of analytes in the liquid phase. Specifically, a higher $E_4/E_6$ ratio indicates a lower molecular weight, a lower degree of aromatic condensation, and a lower degree of humification [25]. As depicted in Fig 1, the $E_4/E_6$ ratio in the supernatant responded differently to various glycine concentrations over the cultivation period. In the Gly0 treatment, the $E_4/E_6$ ratio declined sharply from 4.71 to 1.39 during the initial 28-hour period, then increased slightly to 1.62 by the end of the cultivation period. The Gly0.03 treatment exhibited a decrease in the $E_4/E_6$ ratio from 3.78 to 1.57 within the first 18 hours, followed by fluctuations and an eventual increase to 1.87 thereafter. For the Gly0.06 treatment, the ratio gradually decreased from 3.98 to 1.92 in the first 18 hours, peaked at 2.16 at 28 hours, and then tapered off to 1.80. The Gly0.12 treatment exhibited a steep decline from 6.33 to 2.27 during the initial 18 hours; the ratio then stabilized between 2.42 and 2.55 from 28 to 124 hours, followed by another decrease, and finally settled at 2.11 by the end of the culture period. In the Gly0.24 treatment, the $E_4/E_6$ ratio decreased from 4.05 to 2.13 within the first 6 hours, underwent fluctuations, and eventually increased to 3.77. For the control group

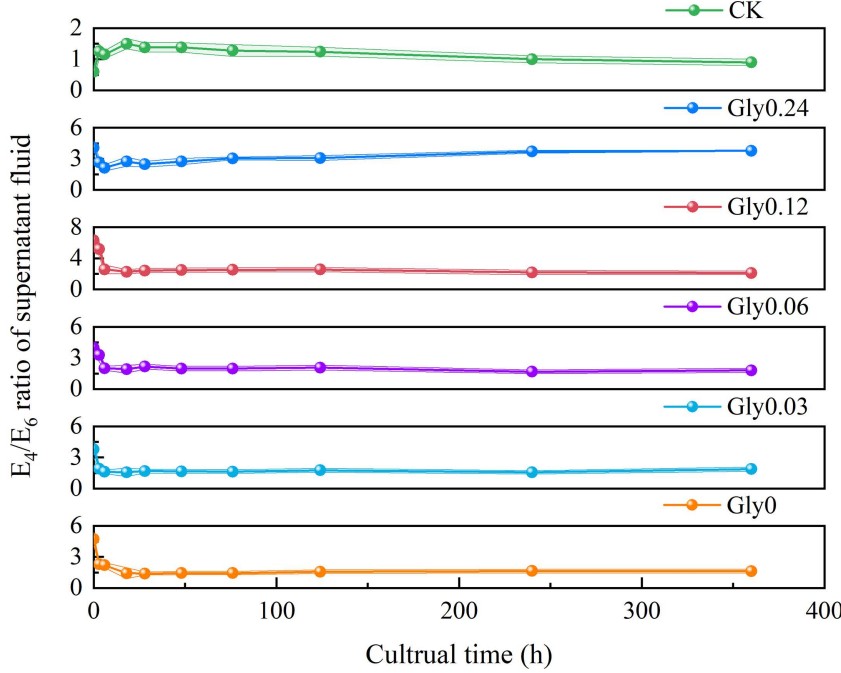

**Fig 1. Effects of glycine at various concentrations on the $E_4/E_6$ ratio in the supernatant.** Treatments supplemented with glycine at concentrations of 0, 0.03, 0.06, 0.12, and 0.24 mol/L are denoted as Gly0, Gly0.03, Gly0.06, Gly0.12 and Gly0.24, respectively. Error bars in the scatter plots represent the standard deviation (SD) for each data point. The same notation applies to all subsequent figures. Detailed SD values for the $E_4/E_6$ ratio at each time point are provided in S1 Table.

(CK), the $E_4/E_6$ ratio first increased then decreased. By the conclusion of the culture period, the percentage reductions in the $E_4/E_6$ ratio relative to the initial values were 65.5%, 50.5%, 54.8%, 66.6%, and 6.8% for the Gly0, Gly0.03, Gly0.06, Gly0.12, and Gly0.24 treatments, respectively. Among all treatments, Gly0.12 exhibited the largest percentage reduction, followed by Gly0, while Gly0.24 showed the smallest reduction (S1 Table).

As shown in Fig 2, throughout the cultivation period, the TOC content in the supernatant under different treatments exhibited similar trends with the addition of the Maillard precursors, including different glycine concentrations. Upon completion of the 360-h cultivation period, the supernatants from the Gly0, Gly0.03, Gly0.06, Gly0.12, Gly0.24 and CK groups showed TOC reductions of 44.9%, 37.0%, 43.5%, 37.9%, 26.0% and 49.8%, respectively. Compared with the CK group, the incorporation of Maillard precursors effectively mitigated TOC loss, among which the Gly0.24 treatment exerted the most pronounced effect in alleviating TOC depletion.

## 3.2. C Content ($C_{HLA}$), $C_{HLA}/C_{FLA}$ ratio and FTIR spectrum of the dark-brown residue

As illustrated in Fig 3, throughout the entire cultivation period, the $C_{HLA}$ content in treatments supplemented with Maillard precursors was significantly higher than that in the CK. The $C_{HLA}$ content in CK exhibited only slight fluctuations, with variations range restricted to 0.02–0.03 g/L. In contrast, all Maillard precursor-supplemented treatments showed a consistent and significant upward trend in $C_{HLA}$ content over time. By the end of cultivation (360 h), the increments in $C_{HLA}$ content relative to the initial measurement (0 h) were as follows: 1295.9% for Gly0, 580.0% for Gly0.03, 1034.6% for Gly0.06, 653.5% for Gly0.12, and 284.8% for Gly0.24 and merely 40.8% for CK. These results clearly demonstrated that the addition of Maillard precursors strongly promoted $C_{HLA}$ accumulation, with the Gly0 treatment achieving the most remarkable enhancement.

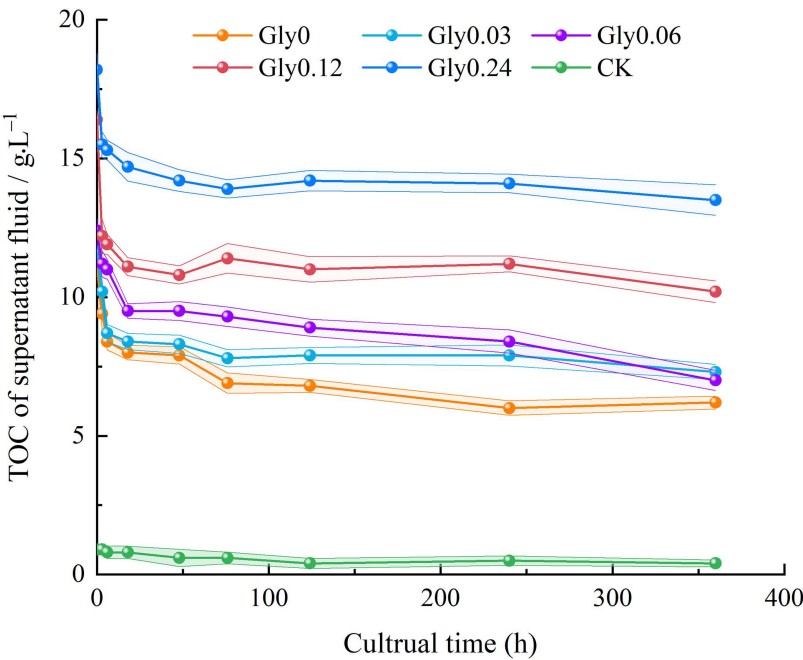

**Fig 2. Effects of glycine at various concentrations on the TOC content in the supernatant.**

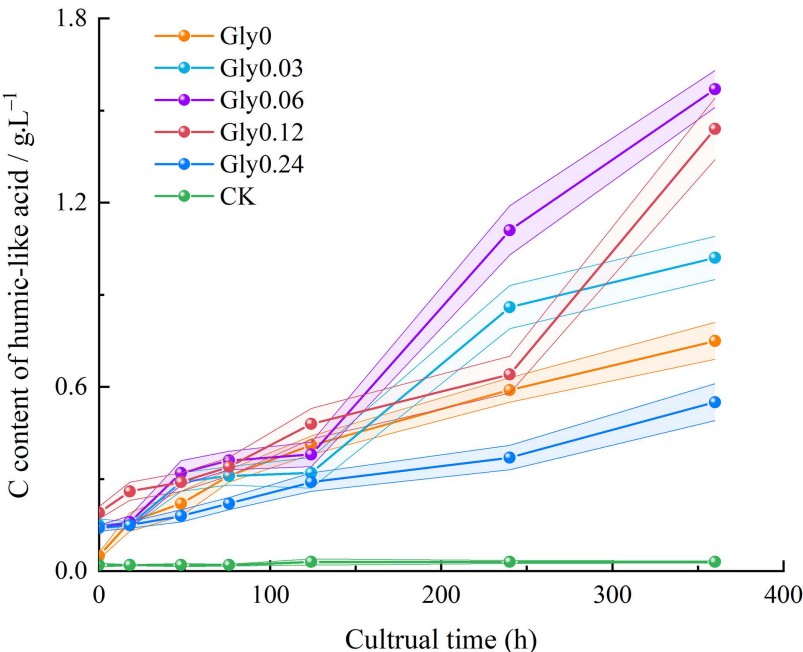

**Fig 3. Effects of glycine at various concentrations on $C_{HLA}$ extracted from the dark-brown residue.**

Alterations in the $C_{HLA}/C_{FLA}$ ratio during liquid shake-flask cultivation were monitored as depicted in Fig 4. The $C_{HLA}/C_{FLA}$ ratios in the Gly0, Gly0.06, Gly0.12 and Gly0.24 treatments displayed an upward trend throughout the cultivation process. In contrast, the Gly0.03 treatment and CK initially exhibited a slight decrease in the $C_{HLA}/C_{FLA}$ ratio during the early stage, followed by a significant rebound in the later stage. At 360 h (the final sampling point), the percentage increases in the $C_{HLA}/C_{FLA}$ ratio relative to the initial value (0 h) were 324.0% (Gly0), 379.1% (Gly0.03), 726.6% (Gly0.06), 781.7% (Gly0.12), 239.0% (Gly0.24) and 56.0% (CK), respectively. Notably, the Gly0.12 treatment achieved the highest increment in the $C_{HLA}/C_{FLA}$ ratio, followed closely by the Gly0.06 treatment. In this study, all treatments containing exogenous Maillard precursors maintained higher $C_{HLA}/C_{FLA}$ ratios compared to CK, which directly confirmed that Maillard precursors could effectively improve the quality of HLSs. Specifically, glycine concentrations of 0.12 and 0.06 mol/L showed the most prominent regulatory effects.

Fig 5 presents the FTIR spectra of the dark-brown residue treated with different glycine concentrations. The broad peak at 3424~3438 cm⁻¹ corresponds to –OH stretching vibrations, which may arise from gibbsite or its interlayer water molecules, or from carboxylic acids or phenols [26]. The intense band at 1622~1635 cm⁻¹ was associated with C=C stretching in aromatic rings [27], while another aromatic C=C stretching peak appeared at 1460~1489 cm⁻¹ [28]. The band at 1385~1389 cm⁻¹ was attributed to COO⁻ group stretching [29], and the 1296~1304 cm⁻¹ region corresponds to C–O–H deformation and C–O stretching in phenolic compounds [30]. Peaks at 1092~1128 cm⁻¹ are assigned to C–O stretching in polysaccharides or polysaccharide-like substances, whereas the sharp absorption peak at 611~656 cm⁻¹ arised from lattice vibration of layered Al–O bonds.

Common features of the FTIR spectra were summarized in Table 1. Relative to the control (CK) the intensity of the broad band at 3424~3438 cm⁻¹ increased to different extents in treatments supplemented with Maillard

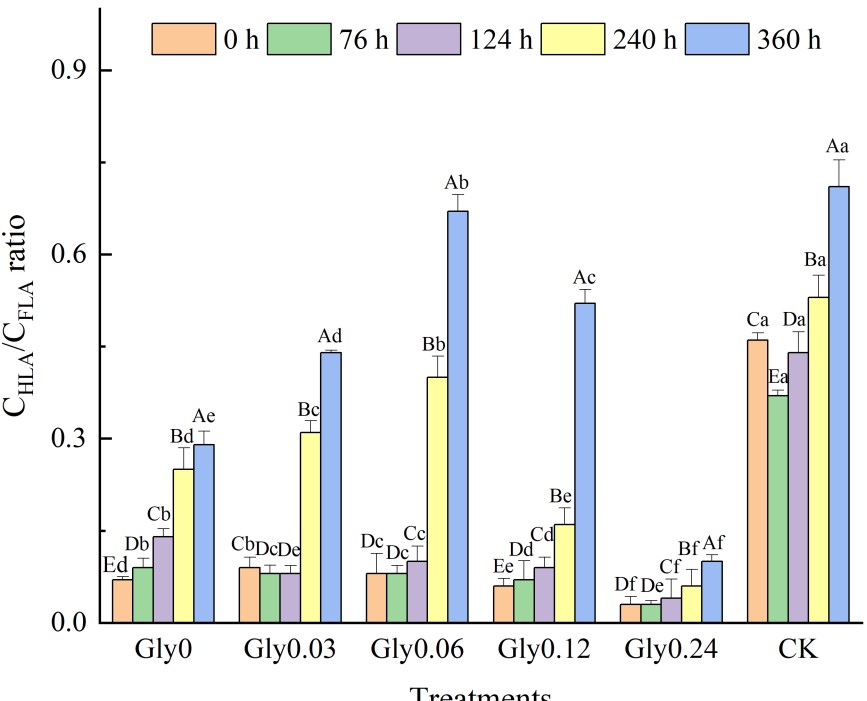

**Fig 4. Effects of glycine at various concentrations on the $C_{HLA}/C_{FLA}$ ratio of the dark-brown residue.** Different capital letters indicate significant differences among different reaction times within the same treatment ($p < 0.05$); different lowercase letters denote significant differences among different treatments under the same cultivation time ($p < 0.05$).

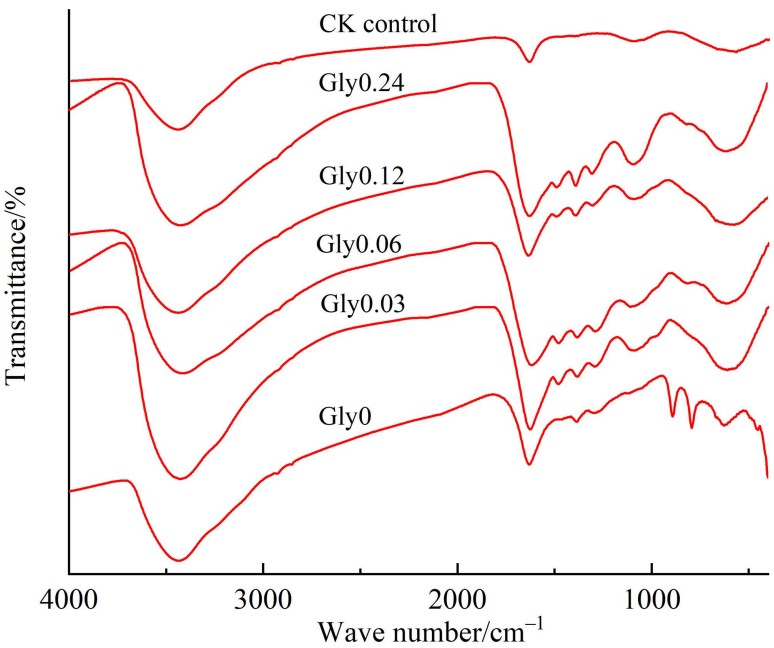

**Fig 5. Effects of glycine at various concentrations on the FTIR spectra of dark-brown residue.**

**Table 1. FTIR relative intensities (% of total area) for the dark-brown residue of Maillard precursors after treatment with different glycine concentrations. Total area was defined as the sum of integrated areas of seven characteristic bands.**

| Wavenumbers (cm⁻¹) Treatments | 3424~3438 | 1622~1635 | 1460~1489 | 1385~1389 | 1296~1304 | 1092~1128 | 611~656 |
|---|---|---|---|---|---|---|---|
| Gly0 | 79.3 | 8.20 | 0.21 | 0.78 | 1.53 | 1.57 | 8.42 |
| Gly0.03 | 80.3 | 8.38 | 0.23 | 0.74 | 0.53 | 1.69 | 8.15 |
| Gly0.06 | 81.1 | 8.83 | 0.35 | 0.85 | 0.86 | 1.69 | 6.28 |
| Gly0.12 | 81.4 | 8.90 | 0.52 | 1.07 | 1.10 | 2.85 | 4.20 |
| Gly0.24 | 79.0 | 8.69 | 0.39 | 0.74 | 0.67 | 3.33 | 7.27 |
| CK | 78.7 | 7.50 | 0.18 | 0.69 | 2.76 | 0.96 | 9.20 |

precursors. Typically, microbial degradation of cellulose, hemicellulose and lignin during composting would reduce the relative intensity of this band [31], suggesting that the observed increase stemmed from the immobilization of organic molecules on gibbsite via hydrogen bonding. This enhanced absorption was likely driven by –OH stretching vibrations from gibbsite (or its interlayer water molecules) and from alcoholic or phenolic hydroxyl groups.

Relative to CK, treatments with Maillard precursors showed increased intensities of absorption peaks at 1622~1635 cm⁻¹, 1460~1489 cm⁻¹, 1385~1389 cm⁻¹ and 1092~1128 cm⁻¹, but decreased intensities at 1296~1304 cm⁻¹ and 611~656 cm⁻¹. This indicated that Maillard precursors increased the degree of aromatization, as well as carboxyl and polysaccharide contents, in the dark-brown residue post-incubation, while reducing phenolic compound and Al–O bond abundance. Among all treatments, Gly0.12 exhibited the highest degree of aromatization and contents of carboxyl and hydroxyl groups, alongside the lowest Al–O bond intensity. In contrast, Gly0.24 resulted in the highest polysaccharide content in the dark-brown residue.

## 3.3. $E_4/E_6$ Ratio and Elemental Composition of HLA

As depicted in Fig 6, the $E_4/E_6$ ratios of HLA extracted from the Gly0, Gly0.03, Gly0.06, Gly0.12 and Gly0.24 treatment groups reached their peaks at 124 h of culture, followed by a gradual decline until the end of the culture period. At the initial stage of culture, the $E_4/E_6$ ratios of the diluted HLA solution across all treatment groups underwent a trend of first increasing and then decreasing—this observation suggested that the HLA molecular structure initially became simplified and subsequently turned more complex. At the end of the culture period (360 h), the $E_4/E_6$ ratios of HLA in the Gly0, Gly0.24, and CK groups increased by 19.1%, 50.7% and 3.4%, respectively, relative to the baseline values at 0 h. In contrast, the $E_4/E_6$ ratios in the Gly0.03, Gly0.06 and Gly0.12 treatment groups decreased by 3.8%, 25.0%, and 38.4%, respectively, compared to the 0 h baseline. Notably, at glycine concentrations of 0.03, 0.06 and 0.12 mol/L, the HLA molecular structure exhibited a tendency to become more complex, among which the highest level of complexity was observed in the Gly0.12 treatment.

HA is predominantly composed of C, H, O, N, and S. The atomic ratios of these elements (e.g., H/C, O/C, and C/N) are well-recognized as robust indicators of HA molecular structure variability and serve as quantitative metrics for evaluating humification degree [32]. Specifically, a lower H/C ratio indicates a higher degree of aromatization (i.e., more condensed benzene ring structures), while the O/C atomic ratio reflects both the humification level and the abundance of O-containing functional groups (e.g., hydroxyl, carboxyl, and carbonyl) in HA. Additionally, the C/N ratio indirectly denotes the content of N-containing compounds (e.g., amines, amides) integrated into HA molecules.

As presented in Table 2, the control group (CK) contained only gibbsite in 0.2 mol/L phosphate buffer (no humification precursors: catechol, glucose, or glycine). Theoretically, no HA could form or be extracted from CK, thus no elemental ratio data were available for this group. The Gly0 treatment (no glycine) was designated as the reference group for comparative analysis of elemental compositions. For the experimental groups (Gly0 to Gly0.24), HLA extracted from the

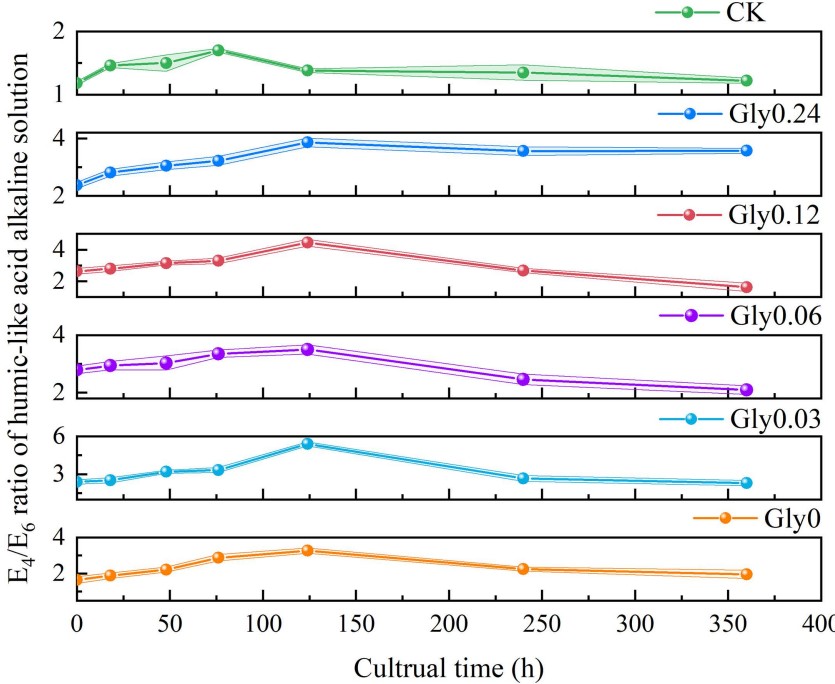

**Fig 6. Effects of glycine at various concentrations on the $E_4/E_6$ ratio of the diluted solution of HLA extracted from dark-brown residue.**

**Table 2. Atomic ratios of HLA molecules extracted from the dark-brown residue following treatment with varying glycine concentrations.**

| Treatments | H/C ratio | C/N ratio | O/C ratio |
|---|---|---|---|
| Gly0 | 1.36±0.02 a | 16.3±0.2 c | 0.91±0.03 d |
| Gly0.03 | 1.25±0.04 b | 16.8±0.4 b | 1.00±0.02 c |
| Gly0.06 | 1.22±0.03 c | 17.0±0.3 a | 1.04±0.03 b |
| Gly0.12 | 1.11±0.01 d | 10.9±0.2 d | 1.38±0.04 a |
| Gly0.24 | 1.39±0.03 a | 8.7±0.1 e | 0.82±0.02 e |

Note: All data are presented as mean±standard deviation (SD, n=3). Different lowercase letters denote significant differences among treatments, determined by one-way analysis of variance (ANOVA) followed by the least significant difference (LSD) test ($p < 0.05$).

dark-brown residue of Gly0.03 (0.03 mol/L glycine), Gly0.06, and Gly0.12 treatments exhibited lower H/C ratios and higher O/C ratios compared to those from Gly0 and Gly0.24. Notably, the HLA from Gly0.12 displayed the highest O/C ratio and the lowest H/C ratio among all groups, indicating that this glycine concentration maximized both aromatic condensation and the accumulation of oxygen-containing functional groups—consistent with the structural insights from FTIR analysis. These results further confirmed that 0.12 mol/L glycine promoted the formation of more mature, functionally diverse HLA in the gibbsite-mediated system. Regarding the C/N ratio, as glycine concentration increased, the C/N ratio of residual HLA first increased gradually and then decreased significantly. When glycine concentrations ranged from 0 to 0.06 mol/L, the content of N-containing compounds in HLA decreased, suggesting that low glycine levels were insufficient to support efficient N incorporation into humic structures (likely due to preferential utilization of carbon precursors for polymerization). In contrast, as glycine concentration increased from 0.06 to 0.24 mol/L, N-containing compounds accumulated in HLA in a dose-dependent manner. Notably, the Gly0.24 treatment exhibited the lowest C/N ratio in HLA molecules, confirming that high glycine concentrations serve as an effective nitrogen source for HLA biosynthesis.

## 4. Discussion

Compared to the CK, the inclusion of Maillard reaction precursors, such as varying concentrations of glycine, led to a range of complexities in the molecular structure of the supernatant after culture, particularly at glycine concentrations of 0 and 0.12 mol/L. These smaller molecules, including phenols, AAs, and sugars, underwent further reactions, polymerizations, and partial polycondensation aided by gibbsite, forming medium- to high-molecular-weight organic substances [33]. The formation of dimers or oligomers through the free radical polymerization of several phenolic species significantly enhanced the abiotic humification process [34]. During this process, organic molecules of varying masses might also form supramolecular bonds, resulting in a humic network [35]. In polycondensation reactions involving humic precursors, nitrogen from AAs could interact with nucleophilic carbons in aromatic rings and carbonyl carbons [10]. Furthermore, glycine could promote humification in integrated catechol-Maillard reaction systems catalyzed by gibbsite, thereby increasing the molecular complexity in the supernatant.

The addition of Maillard precursors (catechol, glucose) mitigated TOC loss in the supernatant, with 0.24 mol/L glycine achieving the highest carbon retention (26.0% reduction). This underscored glycine's role in stabilizing organic C via Al–C complexation. Specifically, gibbsite's surface Al(III) (as a Lewis acid) underwent ligand exchange with the carboxyl group (–COOH) of glycine, forming stable Al–O–C bonds [1]. These bonds prevented the mineralization of organic C by shielding it from hydrolysis; Nkoh et al. [1] similarly reported that Al(III) oxides form strong complexes with amino acid carboxyl groups, reducing TOC loss by 20–30% in humification systems. Additionally, catechol-derived quinones might further cross-link with Al–glycine complexes, forming insoluble aggregates that enhance carbon retention [36]. The incorporation of C-containing precursors and the consequent formation of Al–C complexes, primarily through interaction between gibbsite and precursors like glycine, served to mitigate TOC loss in the supernatant, thereby enhancing carbon sequestration

[37]. In their study on the humification of lignocellulose-like biomass during composting, Zhang et al. [6] noted that employing protein-like precursors proved to be an economical and effective regulatory method for reducing the mineralization of C sources.

During the culture period, the $C_{HLA}$ content in treatments with Maillard precursors was significantly higher than in the CK. The $C_{HLA}$ content in each treatment increased from 0 h to the end of the culture period, with the Gly0 treatment showing the greatest increase, reaching 1295.9%, followed by the Gly0.06 treatment at 1034.6%. The Maillard reaction significantly contributed to HA polymerization [38]. In this study, enhanced $C_{HLA}$ production was observed with the sole presence of catechol and glucose in the Gly0 treatment. Zou et al. [13] confirmed the potential of $MnO_2$ and $O_2$ to form HSs and HA through oxidative polymerization of catechin and glycine without glucose. It could be inferred that polyphenols, primarily catechol, were essential for HLA formation under abiotic humification conditions, aligning with findings by Xing et al. [38]. Polyphenols, typified by catechol, are more labile and reactive than reducing sugars and AAs in this process [36]. They can be oxidized into quinones, which either combine with aromatic AAs to form humus or polymerize with polysaccharides to produce covalent compounds contributing to humus formation [39]. Additionally, the presence of gibbsite ($Al^{3+}$) with Lewis acid functionality could enhance HLA yield [40]. Upon completion of the culture, the $C_{HLA}/C_{FLA}$ ratio in each treatment with Maillard precursors increased, with Gly0.12 showing the largest rise, followed by Gly0.06. Compared to CK, treatments with Maillard precursors increased the $C_{HLA}/C_{FLA}$ ratio, indicating a transformation from simpler molecules (FLA) to more complex molecules (HLA), suggesting increased HLS aromatization and humification [23]. Moore et al. [41] also proposed that simple organic molecules could be geopolymerized into recalcitrant forms by means of the Maillard reaction. During the culturing process, the molecular structure of HLA treated with Maillard precursors evolved from simple to complex. Particularly in the latter half of the culturing period, small molecular substances within the HLAs progressively aggregated to form larger molecular entities. This observation aligned with the findings of Wei et al. [32], who noted that polyphenols and nitrogenous compounds typically polymerize into humus monomer molecules during the second stage of culture. The peak in $E_4/E_6$ ratios of HLA at 124 h (Fig 6) likely stemmed from transient accumulation of low-molecular-weight (LMW) intermediates (e.g., partially polymerized phenols and amino sugar derivatives). At this stage, the rate of LMW intermediate formation (via catechol oxidation and glucose cleavage) exceeded their polycondensation into high-molecular-weight HLA, temporarily increasing $E_4/E_6$ ratios (indicative of lower molecular complexity) [32,38]. As culture proceeded, LMW intermediates were consumed for further polymerization, reducing $E_4/E_6$ ratios and enhancing HLA complexity.

When glycine concentrations were at 0.03, 0.06, and 0.12 mol/L, the structure of the HLA molecules became increasingly complex, incorporating more O-containing functional groups post-culture. This suggested that gibbsite might act as a Lewis acid or oxide, enhancing nucleophilic addition and polycondensation through electron transfer from micromolecular precursors [42], thus increasing the prevalence of O-containing functional groups in HLAs [33]. Conversely, a precursor composition with 0.12 mol/L glycine, 0.06 mol/L catechol, and 0.06 mol/L glucose more effectively promoted the condensation and oxidation of HLA molecules via gibbsite catalysis. A glycine concentration of 0.12 mol/L yielded HLSs with maximal aromaticity (FTIR peak at 1622 cm⁻¹), carboxyl group abundance (1385 cm⁻¹), and O/C ratio (1.38), indicating advanced polycondensation. This aligns with Zhang et al. [43], who reported similar enhancements in Fe(III)-catalyzed systems. Our findings align with Zou et al. [13], who showed metal oxides enhance humic formation, but differ in catalyst mechanism: $MnO_2$ acts as an oxidant (via Mn(IV)/Mn(II) cycles), while gibbsite (Al(III)) is a Lewis acid catalyst—making it suitable for low-redox environments [19,42]. As glycine concentration increased from 0 to 0.24 mol/L, the C/N ratio of HLA molecules extracted from the dark-brown residue first rose gradually and then decreased markedly. With glycine additions from 0 to 0.06 mol/L, the content of N-containing compounds in HLA molecules diminished, suggesting a reduction in the incorporation of these compounds. The ammonia nitrogen content in HLA displayed a negative correlation with glycine levels, possibly due to the transformation of ammonia nitrogen during the abiotic humification process, highlighting its potential role as a significant precursor in HLA formation [44]. However, as the added glycine concentration

extended from 0.06 to 0.24 mol/L, N-containing compounds progressively accumulated in the HLA molecules, particularly under the Gly0.24 treatment, which exhibited the lowest C/N ratio. Elevated glycine (0.24 mol/L) increased N-compound incorporation into HLA (C/N = 8.7), suggesting glycine-derived amines enhance nitrogenous humification. The apparent contradiction in Gly0.24 (high N accumulation but low HLS maturity) arised from glycine concentration-dependent reaction pathways. At 0.24 mol/L, excess glycine preferentially underwent simple nucleophilic addition with catechol (rather than promoting aromatic polycondensation), forming LMW N-containing compounds (e.g., amino-phenols). These LMW compounds had fewer O-containing functional groups (lower O/C ratio) and lower molecular complexity (higher $E_4/E_6$ ratio) [21]. In contrast, 0.12 mol/L glycine balanced N integration and aromatic polymerization: glycine acted as a bridging agent between catechol and glucose, facilitating the formation of high-molecular-weight humic polymers with abundant O-containing groups (e.g., –COOH, –OH) [43]. Recent work by Xing et al. [38] corroborates this, highlighting microbial-independent N-integration pathways. Compared to Zhang et al. [6], who studied biotic-abiotic composting systems, our sterile model isolates abiotic pathways. While composting involves microbial enzyme-driven lignocellulose degradation, our results provide a mechanistic basis for extrapolation: gibbsite could synergize with microbial byproducts (e.g., laccase) to enhance compost humification [10,23].

The addition of Maillard precursors enhanced the aromaticity of molecules, increased the content of alcohol hydroxyl groups, carboxyl groups, and polysaccharides, and decreased the phenolic compounds and Al–O bonds in the dark-brown residue resulting from abiotic humification. •OH production, associated with gibbsite-facilitated humification, attacked phenolic rings to form the aromatic ring skeleton of the dark-brown residue and facilitated ring-opening copolymerization of humic precursors [43]. Edge and Truscott [45] noted that •OH had a superior capacity to promote the oxidative polycondensation of catechol and glycine. The reduction and direct conversion of polyphenol polymers enhanced the humification of the dark-brown residue [34]. Among the various treatments, the Gly0.12 group exhibited the highest aromatization degree and had the most carboxyl and hydroxyl groups in the dark-brown residue, while showing the lowest vibration intensity of the Al–O bonds. When the concentration of polyphenol polymers diminished, with the addition of glycine at a concentration of 0.24 mol/L, the polysaccharide content in the dark-brown residue increased [46]. The optimal glycine concentration (0.12 mol/L) identified here could be translated to lignin-rich waste composting (e.g., straw, wood chips) by adjusting precursor ratios based on waste composition. For example, lignin degradation in straw releases ~0.06 mol/L catechol per kg dry matter, while carbohydrate hydrolysis produced ~0.06 mol/L glucose [6]. Adding glycine at 0.12 mol/L (equivalent to 0.5–1% of dry waste weight) would replicate the optimal catechol: glucose:glycine ratio (1:1:2) from this study. Preliminary trials by Zhang et al. [6] showed that such precursor ratios enhanced HLA formation by 30–40% in straw compost, confirming the practical relevance of our findings.

Glycine interacted with gibbsite and precursors via two key mechanisms: (1) As a bridging molecule, its carboxyl group (–COOH) forms coordinated bonds with Al(III) on gibbsite's Lewis acid sites, while its amino group (–NH$_2$) reacted with glucose-derived carbonyls (Maillard reaction) or catechol-derived quinones (Schiff base formation), anchoring precursors to gibbsite's surface [1,33]. (2) Glycine protonated gibbsite's aluminol groups, increasing surface reactivity for adsorbing and polymerizing precursors [19,42]. This was supported by reduced Al–O bond vibration in Gly0.12 (Table 1), indicating glycine competes for Al(III) coordination sites, suppressing Al–O bonds and promoting Al–C complexation (evidenced by Gly0.24's high TOC retention, Fig 2).

This study had several limitations. First, the sterile shake-flask system represented an idealized environment; natural soils contained microorganisms that might interact with gibbsite and precursors (biotic-abiotic synergy), which was not explored here. Second, only gibbsite was tested as an Al(III) catalyst; other Al-based oxides (e.g., $\gamma$-Al$_2$O$_3$) or mixed metal oxides (e.g., Al–Fe oxides) could exhibit different catalytic efficiencies. Third, the 360-h reaction period only captured short-term humification; long-term (>1 year) stability of HLSs required further investigation. Future work will: (1) introduce soil microorganisms to explore biotic-abiotic synergistic humification; (2) compare gibbsite with other metal oxides to identify optimal catalysts; (3) conduct field trials to validate C sequestration efficiency in real soils; and (4) use liquid

chromatography-mass spectrometry (LC-MS) to characterize HLS molecular composition, enhancing mechanistic under-standing of gibbsite-catalyzed Maillard reactions.

## 5. Conclusions

This study systematically investigated how glycine concentration modulated gibbsite-catalyzed abiotic humification via the Maillard reaction, addressing critical gaps in Al(III) oxide-mediated humification research. Key findings included: (1) 0.24 mol/L glycine maximized TOC retention (26.0% loss) via Al–C complex formation; (2) 0.12 mol/L glycine was optimal, yielding HLSs with maximal aromatic condensation (lowest $E_4/E_6 = 2.11$) and O-containing functional groups (O/C = 1.38); (3) 0.24 mol/L glycine enhanced N-compound accumulation in HLA (C/N = 8.7) but reduced HLS maturity. These results confirmed that 0.06 mol/L catechol, 0.06 mol/L glucose, and 0.12 mol/L glycine balanced aromaticity and functional group diversity for gibbsite-catalyzed humification. Limitations of this study included the sterile lab-scale system (which excludes microbial contributions to humification) and the use of a single Al oxide. Future research should: (1) explore bio-abiotic synergy by incorporating compost-derived microorganisms; (2) test mixed metal oxides (e.g., gibbsite+$MnO_2$) to leverage complementary catalytic properties; (3) validate optimal precursor ratios in pilot-scale composting of lignin-rich wastes (e.g., corn stover). These advancements will further enhance the application of gibbsite-mediated humification in carbon sequestration and waste valorization.

## Supporting information

**S1 Table.** $E_4/E_6$ ratios and SD values of supernatant at different culture times.
(DOCX)

## Author contributions

**Conceptualization:** Kai Li, Shuai Wang.

**Data curation:** Kai Li, Donghui Dai, Haoyu Gao.

**Funding acquisition:** Kai Li, Shuai Wang.

**Investigation:** Qi Han, Donghui Dai, Mingshuo Wang, Haihang Sun.

**Visualization:** Jingjing Wang, Jingwei Gao.

**Writing – original draft:** Kai Li.

**Writing – review & editing:** Shuai Wang.

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
