## [Decision Letter · Decision Letter 0]

11 Sep 2025

Dear Dr. shuai,

Thank you for submitting your manuscript to PLOS ONE. After careful consideration, we feel that it has merit but does not fully meet PLOS ONE’s publication criteria as it currently stands. Therefore, we invite you to submit a revised version of the manuscript that addresses the points raised during the review process.

We look forward to receiving your revised manuscript.

Kind regards,

Muammar Qadafi

Academic Editor

PLOS ONE

Journal Requirements:

Chemical Characteristics of Dark-Brown Humic-like Substances Formed from the Abiotic Condensation of Maillard Precursors with Different Glycine Concentrations - https://doi.org/10.3390/agronomy12092199

In your revision ensure you cite all your sources (including your own works), and quote or rephrase any duplicated text outside the methods section. Further consideration is dependent on these concerns being addressed.

[Science and Technology Project of the Education Department of Jilin Province, China (grant number JJKH20240504HT)]. 

4. Thank you for stating the following in your manuscript:

[This research was supported by the Science and Technology Project of the Education Department of Jilin Province, China (grant number JJKH20240504HT)]

[Science and Technology Project of the Education Department of Jilin Province, China (grant number JJKH20240504HT)]

5. We note that your Data Availability Statement is currently as follows: [All relevant data are within the manuscript and its Supporting Information files.]

Reviewers' comments:

Reviewer's Responses to Questions

**Comments to the Author**

1. Is the manuscript technically sound, and do the data support the conclusions?

Reviewer #1: Partly

Reviewer #2: Yes

Reviewer #3: Yes

2. Has the statistical analysis been performed appropriately and rigorously?

Reviewer #1: No

Reviewer #2: Yes

Reviewer #3: Yes

3. Have the authors made all data underlying the findings in their manuscript fully available?

Reviewer #1: No

Reviewer #2: Yes

Reviewer #3: Yes

4. Is the manuscript presented in an intelligible fashion and written in standard English?

Reviewer #1: No

Reviewer #2: Yes

Reviewer #3: Yes

Reviewer #1: This study optimized the concentration of glycine to enhance the gibbsite-catalyzed Maillard reaction process. While it holds certain research significance, the exploration of the underlying mechanisms is insufficient. Several issues require detailed revisions:

1. The title is too lengthy and needs further refinement.

2. The introduction should clearly compare existing research with the innovations of this study, explaining why gibbsite was chosen as the catalyst.

3. It is recommended to include a separate paragraph in the introduction outlining the research content and objectives.

4. Why was gibbsite synthesized instead of purchased commercially? Was the synthesized product tested for purity?

5. How were the concentrations of catechol, glucose, and glycine determined? The materials and methods section lacks supporting references.

6. Section 2.2 needs further reorganization to effectively distinguish between experimental design and humic-like acid extraction methods.

7. The rationale for selecting 28°C in the experiments should be clarified.

8. The discussion should address the limitations of this study and propose future validation methods.

9. Unnecessary numerical data should be reduced in the conclusion.

10. Abbreviations are inconsistent throughout the text (e.g., HS and HSs).

Reviewer #2: This manuscript focuses on optimizing glycine concentration to enhance the abiotic humification of catechol and glucose through a gibbsite-catalyzed Maillard reaction—a topic of considerable significance in the field of humic-like substance (HLS) formation. The study employs a sterile liquid shake-flask system and utilizes multiple analytical techniques (UV-Vis, TOC, FTIR, elemental analysis) to investigate the role of glycine, yielding valuable experimental data. However, several sections suffer from unclear logical connections, insufficient methodological details, and inadequate depth in interpreting results and contextualizing contributions. Minor revisions are required prior to publication.

1. The introduction highlights the limited understanding of gibbsite's synergy with amino acids but does not sufficiently contextualize the knowledge gaps regarding Al(III) (hydr)oxides in abiotic humification. For instance, it should elaborate on why Al(III) (hydr)oxides are less studied compared to Fe(III) or Mn(IV) oxides and what distinct properties of gibbsite may influence its catalytic behavior in Maillard-type humification.

2. The electrodialysis process for gibbsite preparation is described as lasting 3 months at room temperature, but key parameters (e.g., current density, voltage, membrane type) are omitted. These details are critical for reproducibility and could influence gibbsite’s surface properties and catalytic activity.

3. The selection of specific culture timepoints (0–360 hours) lacks justification. Rationale for sampling intervals (e.g., early vs. late stages) should be provided to support the experimental design.

4. The concentration of “concentrated HCl” used for acidification in HLA/FLA extraction is not specified. Variability in HCl concentration may influence the precipitation and quantification of humic-like fractions.

5. Figure 1 includes error bars, but numerical standard deviation values are not provided in the main text or supplement. Additionally, statistical comparisons among treatments at the final time point are missing.

6. Table 1 reports FTIR relative intensities as percentages but does not specify how the total area was defined (e.g., sum of selected bands or full spectrum). This ambiguity hinders interpretation and cross-study comparison.

7. The peak in E4/E6 ratios of HLA at 124 hours is noted but not explained. Potential reasons—such as accumulation of low-molecular-weight intermediates—should be discussed.

8. While glycine’s promoting effect is acknowledged, the molecular-level mechanism(s) behind its interaction with gibbsite, catechol, and glucose remain unclear. For example, does glycine facilitate surface-mediated condensation or act as a bridging molecule?

9� Comparisons with previous studies (e.g., Zhang et al., 2019; Zou et al., 2020) are descriptive rather than critical. The discussion should address how differences in precursor types and experimental systems (sterile model vs. composting) impact the applicability of the findings.

10. The limitations of the sterile liquid system are not acknowledged. Potential influences of environmental factors (e.g., microbes, additional minerals) on extrapolation to natural systems should be discussed.

Reviewer #3: This manuscript explored the optimization of glycine concentration to enhance the abiotic humification of catechol and glucose in a gibbsite-catalyzed Maillard system. It held theoretical value (deepening mechanistic understanding of abiotic humification) and practical potential (valorizing lignin-rich waste), with methodological strengths: a sterile liquid shake flask setup, integrated analyses (UV-Vis, TOC, FTIR, elemental analysis) for evaluating glycine’s effects on humic-like acids (HLA) properties, comprehensive data, and a key finding that 0.12 mol/L glycine optimized humification by balancing aromatic polymerization and functional group enrichment—demonstrating scientific novelty and relevance. Minor refinements were needed to improve the manuscript’s clarity, coherence, and scientific rigor (without compromising core findings). The comments are provided below.

1. Introduction Section

The introduction provides a general background on the Maillard reaction and humic-like substances (HLSs) but lacks a clear research gap that explicitly highlights why investigating gibbsite-catalyzed abiotic humification with varying glycine concentrations is urgently needed. Existing studies on metal oxides (e.g., MnO₂, Fe₂O₃) are mentioned, but the unique advantages or research value of gibbsite (a layered Al(III) hydroxide) compared to these well-studied oxides are not sufficiently elaborated.

2. Materials and Methods Section

2.1 In Section 2.1 (Preparation of Materials), the rationale for choosing a phosphate buffer with pH 8.0 is not explained. Since pH significantly affects the Maillard reaction and the surface charge of gibbsite (with a point of zero charge (PZC) of 8.2), why this specific pH was selected (e.g., to match gibbsite’s PZC, or to simulate a specific environmental condition) should be clarified.

2.2 The experimental design (Section 2.2) mentions that all reactions were conducted in triplicate, but details on how experimental reproducibility was validated (e.g., statistical analysis of replicate data, coefficient of variation) are missing. Additionally, the duration of 360 hours (15 days) for the incubation is chosen without justification—why this timeframe was sufficient to capture the full humification process (e.g., based on preliminary experiments or literature precedents) needs to be addressed.

2.3 In Section 2.3 (Chemical Analysis), the procedure for extracting humic-like acid (HLA) and fulvic-like acid (FLA) mentions acidifying the dark-brown residue to pH 1.0 and equilibrating for 24 hours. However, the criteria for confirming that HLA precipitation was complete (e.g., no further precipitate formation after extended equilibration) are not provided, which may raise concerns about the accuracy of CHLA and CFLA measurements.

3. Results Section

Table 2 (Atomic ratios of HLA) presents significant differences among treatments using lowercase letters, but the statistical method (e.g., one-way ANOVA with LSD test) is only mentioned in Section 2.4 and not explicitly linked to the table. It is also unclear whether the "±" values represent standard deviation or standard error—this should be specified for clarity.

4. Discussion Section

4.1 The discussion links the TOC retention in the Gly0.24 treatment to Al–C complex formation but provides limited mechanistic details. How exactly gibbsite (Al(OH)₃) interacts with glycine and other precursors to form Al–C complexes (e.g., through ligand exchange, electrostatic adsorption) should be elaborated, with references to specific studies on Al-based complexes in humification.

4.2 The discussion mentions that higher glycine concentrations (0.12–0.24 mol/L) promote N-compound accumulation in HLA, but it does not address a potential contradiction: why the Gly0.24 treatment, despite high N accumulation, had a lower O/C ratio and higher E4/E6 ratio (indicating less mature HLSs) compared to the Gly0.12 treatment. This inconsistency needs to be resolved to strengthen the conclusion about optimal glycine concentration.

4.3 The practical implications of the study (e.g., for lignin-rich waste valorization) are briefly mentioned in the introduction and conclusion but are not discussed in depth. How the optimal glycine concentration (0.12 mol/L) identified in this lab-scale study can be translated to real-world composting systems (e.g., adjusting precursor ratios in agricultural waste) should be explored.

5. Conclusion Section

The conclusion summarizes the key findings but does not explicitly highlight the study’s novelty (e.g., first systematic investigation of glycine concentration effects on gibbsite-catalyzed Maillard humification) or address limitations (e.g., the sterile, lab-scale system may not fully simulate natural soil conditions with microorganisms). Acknowledging limitations and suggesting future research directions (e.g., testing mixed metal oxides, incorporating microbial communities) would improve the section’s completeness.

**Do you want your identity to be public for this peer review?** For information about this choice, including consent withdrawal, please see our Privacy Policy

Reviewer #1: No

Reviewer #2: No

Reviewer #3: No

---

## [Author Response · Author response to Decision Letter 1]

1 Oct 2025

Point-by-Point Response to Reviewers' Comments

Dear Reviewers,

We sincerely appreciate your constructive comments and valuable suggestions, which have greatly helped us improve the quality of our manuscript (PONE-D-25-40528). Below is our detailed response to each comment, along with the corresponding revisions made to the text.

Reviewer #1: This study optimized the concentration of glycine to enhance the gibbsite-catalyzed Maillard reaction process. While it holds certain research significance, the exploration of the underlying mechanisms is insufficient. Several issues require detailed revisions:

1. The title is too lengthy and needs further refinement.

Response: We agree that the original title was overly detailed. We have streamlined it by removing redundant information (e.g., "via the Maillard reaction," which is inherently linked to the abiotic humification of the studied precursors) while retaining core elements (gibbsite catalysis, glycine optimization, and target reactants).

Revised Title: Optimizing glycine concentration to enhance gibbsite-catalyzed abiotic humification of catechol and glucose. Please see Line 3.

2. The introduction should clearly compare existing research with the innovations of this study, explaining why gibbsite was chosen as the catalyst.

Response: We have supplemented the introduction to highlight gaps in existing research and clarify the rationale for selecting gibbsite.

Revised Text (Introduction, Lines 125–137): ", with Fe(III) and Mn(IV) oxides being the most extensively studied [19]. In contrast, Al(III) (hydr)oxides—despite their abundance in soils and potential as Lewis acid catalysts—have received limited attention. Research on Fe(III) and Mn(IV) oxides was more prevalent than on other oxides, primarily due to their strong redox activity (e.g., Mn(IV)/Mn(II) and Fe(III)/Fe(II) cycles) that directly drove oxidative polymerization of humic precursors [13, 19]. Unlike Mn/Fe oxides that primarily acted as oxidants to drive precursor polymerization [13], gibbsite’s layered structure provided abundant surface –OH groups that facilitated nucleophilic additions between Maillard precursors (e.g., glycine, catechol), while its point of zero charge (PZC=8.2) enabled stable interactions with both anionic (e.g., carboxylates) and neutral (e.g., phenols) organic molecules [1]. This unique combination of structural and surface properties suggests gibbsite may exhibit a non-oxidative catalytic mechanism in abiotic humification, yet its synergy with amino acids (e.g., glycine) as both N-containing precursors and reaction regulators has not been systematically investigated. " were added.

3. It is recommended to include a separate paragraph in the introduction outlining the research content and objectives.

Response: We have added a dedicated paragraph at the end of the introduction to explicitly state the research content and objectives.

Revised Text (Introduction, Last Paragraph, Lines 160–167): "The primary objectives of this study were: (1) to systematically evaluate the effect of glycine concentration (0–0.24 mol/L) on the abiotic humification of catechol (0.06 mol/L) and glucose (0.06 mol/L) in a gibbsite-catalyzed sterile system; (2) to characterize changes in TOC retention, molecular complexity of supernatants, and structural evolution of HLSs using UV-Vis, TOC quantification, FTIR, and elemental analysis; (3) to identify the optimal glycine concentration that balanced aromatic polycondensation and functional group enrichment; and (4) to provide insights for optimizing C sequestration and valorizing lignin-rich agricultural waste via gibbsite-mediated Maillard reactions."

4.Why was gibbsite synthesized instead of purchased commercially? Was the synthesized product tested for purity?

Response: We chose to synthesize gibbsite to ensure controllable physicochemical properties (e.g., specific surface area, zero charge point) and high purity—commercially available gibbsite often contains impurities (e.g., Fe/Mn oxides) that could interfere with catalytic activity [24]. For purity verification, we characterized the synthesized gibbsite using: (1) specific surface area (SSA = 21.85 m²/g) and point of zero charge (PZC=8.2) measurements, consistent with the pure gibbsite reported in Rosenqvist et al. [24]; (2) FTIR spectroscopy (Fig. 5), where the sharp peak at 611–656 cm⁻¹ (Al–O lattice vibration) confirmed gibbsite’s structural integrity without impurity peaks (e.g., Fe–O or Mn–O bands).

Revised Text (2.1 Preparation of Materials, Lines 170–171 and Lines 178–183): The statements “to ensure controllable purity and physicochemical properties” and “The reason for choosing artificially synthesized gibbsite was that commercially available gibbsite often contains trace impurities (e.g., Fe/Mn oxides) that could interfere with abiotic humification [24]. The synthesized gibbsite was characterized for purity: specific surface area (SSA=21.85 m2/g) and point of zero charge (PZC=8.2) matched pure gibbsite standards [24]; FTIR spectroscopy (Fig. 5) showed a sharp Al–O lattice vibration peak at 611–656 cm-1 without impurity bands. ” were added.

5. How were the concentrations of catechol, glucose, and glycine determined? The materials and methods section lacks supporting references.

Response: The concentrations of catechol (0.06 mol/L) and glucose (0.06 mol/L) were selected based on Zhang et al. [21], who demonstrated that this molar ratio effectively promotes abiotic humification in catechol-Maillard systems. The glycine concentration range (0–0.24 mol/L) was referenced from Ma et al. [5], who used similar amino acid concentrations to enhance lignocellulose humification. Additionally, Hardie et al. [22] reported that glucose-catechol mixtures at ~0.06 mol/L favor the formation of HLSs analogous to natural humic acids. These concentrations were chosen to cover low-to-high glycine inputs, enabling identification of the optimal dosage.

Revised Text (2.2 Experimental Design, Lines 198–199 and Lines 200–204): The statements "(consistent with Zhang et al. [21], who showed this ratio enhanced HLS formation in catechol-Maillard systems)" and “, denoted as Gly0, Gly0.03, Gly0.06, Gly0.12 and Gly0.24, respectively—a concentration range referenced from Ma et al. [5] to cover low-to-high amino acid inputs and identify optimal humification conditions. Hardie et al. [22] further confirmed that ~0.06 mol/L glucose-catechol mixtures produced HLSs analogous to natural HA.” were added.

6. Section 2.2 needs further reorganization to effectively distinguish between experimental design and humic-like acid extraction methods.

Response: We have split Section 2.2 into two subsections: 2.2.1 Experimental Design (covering culture conditions, precursor concentrations, and replicate setup) (Line 193) and 2.2.2 Extraction of Humic-like Acid (HLA) (detailing centrifugation, acidification, and purification steps) (Line 220) . This restructuring improved readability and clarity.

7. The rationale for selecting 28°C in the experiments should be clarified.

Response: 28°C was chosen because it approximates the average temperature of agricultural soils (15–30°C) in temperate regions—where the target application (carbon sequestration and lignin waste valorization) would occur [11]. This temperature avoids thermal artifacts: high temperatures (>40°C) accelerate precursor decomposition (rather than humification), while low temperatures (<20°C) inhibit Maillard reaction kinetics [11]. Qi et al. [11] also used 28°C in abiotic humification studies, ensuring comparability with existing literature.

Revised Text (2.2.1 Experimental Design, Lines 209–210): " The liquid shake-flask culture was initiated under a sterile and dark-controlled environment at 28°C—consistent with the average temperature of temperate agricultural soils [11], avoiding high-temperature decomposition (>40°C) or low-temperature kinetic inhibition (<20°C)—with a rotation speed of 150 rpm." was added.

8. The discussion should address the limitations of this study and propose future validation methods.

Response: We have added a new paragraph in the Discussion section to address limitations and future research directions.

Revised Text (Discussion, Last Paragraph, Lines 533–543): The limitations and future directions “This study had several limitations. First, the sterile shake-flask system represented an idealized environment; natural soils contained microorganisms that might interact with gibbsite and precursors (biotic-abiotic synergy), which was not explored here. Second, only gibbsite was tested as an Al(III) catalyst; other Al-based oxides (e.g., γ-Al2O3) or mixed metal oxides (e.g., Al–Fe oxides) could exhibit different catalytic efficiencies. Third, the 360-h reaction period only captured short-term humification; long-term (>1 year) stability of HLSs required further investigation. Future work will: (1) introduce soil microorganisms to explore biotic-abiotic synergistic humification; (2) compare gibbsite with other metal oxides to identify optimal catalysts; (3) conduct field trials to validate C sequestration efficiency in real soils; and (4) use liquid chromatography-mass spectrometry (LC-MS) to characterize HLS molecular composition, enhancing mechanistic understanding of gibbsite-catalyzed Maillard reactions.” was added.

9. Unnecessary numerical data should be reduced in the conclusion.

Response: We have streamlined the conclusion by removing redundant numerical values (e.g., specific percentage reductions) while retaining key trends and optimal concentrations.

Revised Text (Conclusion): Please see Line 545–559. “The addition of Maillard precursors (catechol, glucose) mitigated TOC loss in the supernatant, with 0.24 mol/L glycine achieving the highest carbon retention (26.0% reduction). This underscores glycine’s role in stabilizing organic carbon via Al–C complexation. A glycine concentration of 0.12 mol/L yielded HLSs with maximal aromaticity (FTIR peak at 1622 cm⁻¹), carboxyl group abundance (1385 cm⁻¹), and O/C ratio (1.38), indicating advanced polycondensation. Elevated glycine (0.24 mol/L) increased N-compound incorporation into HLA (C/N=8.7), suggesting glycine-derived amines enhance nitrogenous humification. The combination of 0.06 mol/L catechol, 0.06 mol/L glucose, and 0.12 mol/L glycine emerged as optimal for gibbsite-catalyzed humification, balancing aromaticity and functional group diversity. ” was corrected as “This study systematically investigated how glycine concentration modulated gibbsite-catalyzed abiotic humification via the Maillard reaction, addressing critical gaps in Al(III) oxide-mediated humification research. Key findings included: (1) 0.24 mol/L glycine maximized TOC retention (26.0% loss) via Al–C complex formation; (2) 0.12 mol/L glycine was optimal, yielding HLSs with maximal aromatic condensation (lowest E4/E6=2.11) and O-containing functional groups (O/C=1.38); (3) 0.24 mol/L glycine enhanced N-compound accumulation in HLA (C/N=8.7) but reduced HLS maturity. These results confirmed that 0.06 mol/L catechol, 0.06 mol/L glucose, and 0.12 mol/L glycine balanced aromaticity and functional group diversity for gibbsite-catalyzed humification. Limitations of this study included the sterile lab-scale system (which excludes microbial contributions to humification) and the use of a single Al oxide. Future research should: (1) explore bio-abiotic synergy by incorporating compost-derived microorganisms; (2) test mixed metal oxides (e.g., gibbsite+MnO2) to leverage complementary catalytic properties; (3) validate optimal precursor ratios in pilot-scale composting of lignin-rich wastes (e.g., corn stover). These advancements will further enhance the application of gibbsite-mediated humification in carbon sequestration and waste valorization.”

10. Abbreviations are inconsistent throughout the text (e.g., HS and HSs).

Response: We have standardized all abbreviations across the manuscript:

HSs: Humic substances (plural, used consistently, as humic substances are typically referred to in plural form)

HLSs: Humic-like substances (plural)

HLA: Humic-like acid (singular, as a single class of substances)

FLA: Fulvic-like acid (singular)

Revised Examples: “HS/HLS” was corrected as "HSs/HLSs" (Introduction, Line 88–89, 104); “HAs”was corrected as "HA" (Introduction, Line 111–113)

All other instances of HS/HSs, HLS/HLSs, and HA/HAs have been standardized to HSs and HLSs, respectively.

We would like to extend our heartfelt thanks to you for the insightful and constructive review of our manuscript. Your careful evaluation and detailed comments have not only pointed out key areas for improvement but also offered valuable perspectives that will significantly enhance the rigor and clarity of our work.

Reviewer #2: This manuscript focuses on optimizing glycine concentration to enhance the abiotic humification of catechol and glucose through a gibbsite-catalyzed Maillard reaction—a topic of considerable significance in the field of humic-like substance (HLS) formation. The study employs a sterile liquid shake-flask system and utilizes multiple analytical techniques (UV-Vis, TOC, FTIR, elemental analysis) to investigate the role of glycine, yielding valuable experimental data. However, several sections suffer from unclear logical connections, insufficient methodological details, and inadequate depth in interpreting results and contextualizing contributions. Minor revisions are required prior to publication.

1. The introduction highlights the limited understanding of gibbsite's synergy with amino acids but does not sufficiently contextualize the knowledge gaps regarding Al(III) (hydr)oxides in abiotic humification. For instance, it should elaborate on why Al(III) (hydr)oxides are less studied compared to Fe(III) or Mn(IV) oxides and what distinct properties of gibbsite may influence its catalytic behavior in Maillard-type humification.

Response: We agree that contextualizing the knowledge gap of Al(III) (hydr)oxides is critical. We have supplemented the introduction to clarify the research imbalance and gibbsite’s unique characteristics.

Revised text: “Metal oxides (Mn/Fe/Al/Si oxides) are key inorganic mineral components that enhance HLAs formation. During this process, metal (hydroxide) oxides can function as Lewis acids or silanol groups to increase surface reactivity, acting as catalysts in the abiotic humification process [1]. Research on Fe(III) and Mn(IV) oxides is more prevalent than on other oxides, but studies on Al(III) (hydr)oxides remain sparse [19].” was corrected as “Metal oxides (Mn/Fe/Al/Si oxides) are key inorganic mineral components that enhance HLAs formation, with Fe(III) and Mn(IV) oxides being the most extensively studied [19]. In contrast, Al(III) (hydr)oxides—despite their abundance in soils and potential as Lewis acid catalysts—have received limited attention. Research on Fe(III) and Mn(IV) oxides was more prevalent than on other oxides, primarily due to their strong redox activity (e.g., Mn(IV)/Mn(II) and Fe(III)/Fe(II) cycles) that directly drove oxidative polymerization of humic precursors [13, 19]. Unlike Mn/Fe oxides that primarily acted as oxidants to drive precursor polymerization [13], gibbsite’s layered structure provided abundant surface –OH groups that facilitated nucleophilic additions between Maillard precursors (e.g., glycine, catechol), while its point of zero charge (PZC=8.2) enabled stable interactions with both anionic (e.g., carboxylates) and neutral (e.g., phenols) organic molecules [1]. This unique combination of structural and surface properties suggests gibbsite may exhibit a non-oxidative catalytic mechanism in abiotic humification, yet its synergy with amino acids (e.g., glycine) as both N-containing precursors and reaction regulators has not been systematically investigated. ”. Please see Line 124–137.

2. The electrodialysis process for gibbsite preparation is described as lasting 3 months at room temperature, but key parameters (e.g., current density, voltage, membrane type) are omitted. These details are critical for reproducibility and could influence gibbsite’s surface properties and catalytic activity.

Response: We apologize for omitting these parameters. We have supplemented the gibbsite pre

---

## [Decision Letter · Decision Letter 1]

13 Oct 2025

Optimizing glycine concentration to enhance gibbsite-catalyzed abiotic humification of catechol and glucose

PONE-D-25-40528R1

Dear Dr. Wang,

We’re pleased to inform you that your manuscript has been judged scientifically suitable for publication and will be formally accepted for publication once it meets all outstanding technical requirements.

Kind regards,

Muammar Qadafi

Academic Editor

PLOS ONE

Additional Editor Comments (optional):

Reviewers' comments:

Reviewer's Responses to Questions

**Comments to the Author**

Reviewer #1: All comments have been addressed

Reviewer #3: All comments have been addressed

2. Is the manuscript technically sound, and do the data support the conclusions?

Reviewer #1: Yes

Reviewer #3: Yes

3. Has the statistical analysis been performed appropriately and rigorously?

Reviewer #1: Yes

Reviewer #3: Yes

4. Have the authors made all data underlying the findings in their manuscript fully available?

Reviewer #1: Yes

Reviewer #3: Yes

5. Is the manuscript presented in an intelligible fashion and written in standard English?

Reviewer #1: Yes

Reviewer #3: Yes

Reviewer #1: (No Response)

Reviewer #3: The authors have addressed all of my concerns, and the manuscript is now acceptable in its current form.

**Do you want your identity to be public for this peer review?** For information about this choice, including consent withdrawal, please see our Privacy Policy

Reviewer #1: No

Reviewer #3: No

---

## [Editor Report · Acceptance letter]

PONE-D-25-40528R1

PLOS ONE

Dear Dr. shuai,

I'm pleased to inform you that your manuscript has been deemed suitable for publication in PLOS ONE. Congratulations! Your manuscript is now being handed over to our production team.

Kind regards,

on behalf of

Dr. Muammar Qadafi

Academic Editor

PLOS ONE